# NEWTON LOSSES: USING CURVATURE INFORMATION FOR LEARNING WITH DIFFERENTIABLE ALGORITHMS

## ABSTRACT

In many supervised learning problems, model predictions are compared to ground truth labels using simple convex losses such as softmax cross-entropy and squared error. In weakly-supervised learning, more complex losses involving problem-specific algorithmic procedures and knowledge, e.g., differentiable shortest-path computations or sorting algorithms, are common. These losses can be hard to optimize as they are non-convex in the model output and may exhibit vanishing and exploding gradients. To alleviate this issue, we present Newton Losses, a method for boosting the performance of non-convex and hard to optimize losses by locally approximating an existing loss function with a quadratic (incorporating second-order information). As Newton Losses only replaces the loss function, the method allows training the neural network with gradient descent and is computationally efficient. We apply Newton Losses to eight differentiable algorithm methods for the multi-digit MNIST sorting benchmark and the Warcraft shortest-path benchmark.

## 1 INTRODUCTION

Traditionally, fully-supervised classification and regression learning relies on convex loss functions such as MSE or cross-entropy, which are easy-to-optimize by themselves. However, the need for large amounts of ground truth annotations is a limitation of fully-supervised learning; thus, weakly-supervised learning has gained popularity. Rather than using fully annotated data, weakly-supervised learning utilizes problem-specific algorithmic knowledge incorporated into the loss function. For example, instead of supervising ground truth values, supervision can be given in the form of ordering information. However, incorporating such knowledge into the loss can make it difficult to optimize, e.g., by making the loss non-convex in the model output, non-monotonic, introducing bad local minima, vanishing gradients, and/or exploding gradients, negatively impacting training [1], [2].

Loss functions that integrate problem-specific knowledge can range from rather simple contrastive losses [3] to rather complex losses that require the integration of differentiable algorithms [4]. In this work, we primarily focus on the (harder) latter category, which allows for solving specialized tasks such as inverse rendering [5]–[7], learning-to-rank [2], [8]–[13], self-supervised learning [14], differentiation of optimizers [15], [16], and top-k supervision [10], [12], [17]. In this paper, we summarize these loss functions under the umbrella of algorithmic losses [18] as they introduce algorithmic knowledge into the training objective.

While the success of neural network training is primarily due to the backpropagation algorithm and stochastic gradient descent (SGD), there has also been a strong line of work on second-order optimization for neural network training [19]–[28]. Compared to first-order methods like SGD, second-order optimization methods exhibit improved convergence rates and therefore require fewer training steps; however, they have two major limitations [26], namely (i) computing the inverse of the curvature matrix for a large and deep neural network is computationally substantially more expensive than simply computing the gradient with backpropagation, which makes second-order methods practically inapplicable in many cases [29]; (ii) networks trained with second-order information exhibit reduced generalization capabilities [30].

Inspired by ideas from second-order optimization, in this work, we propose a novel method for incorporating second-order information into training with non-convex and hard to optimize algorithmic losses. Loss functions are usually cheaper to evaluate than a neural network. Further, loss functions operate on lower dimensional spaces than the parameters of neural networks. If the loss function

becomes the bottleneck in the optimization process because it is difficult to optimize, it suggests itself to use a stronger optimization method that requires fewer steps like second-order optimization. However, as applying second-order methods to neural networks is expensive and limits generalization, we want to train the neural network with first-order SGD. Therefore, we propose Newton Losses, a method for locally approximating loss functions with a quadratic via their second-order information. Thereby, Newton Losses provides a (locally) convex loss leading to better optimization behavior, while training the actual neural network with gradient descent. To clarify the notation, we note that, with *"Newton Losses"*, we refer to the *principle of converting an existing loss into a Newton loss.*

For the quadratic approximation of the algorithmic losses, we propose two variants of Newton Losses: (i) *Hessian-based Newton Losses*, which comprises a generally stronger method but requires an estimate of the Hessian. Depending on the choice of differentiable algorithm, choice of relaxation, or its implementation, the Hessian may, however, not be available. Thus, we also relax the method to (ii) *empirical Fisher-based Newton Losses*, which derive the curvature information from the empirical Fisher matrix, which only depends on the gradients. The Fisher variant can be easily implemented on top of existing algorithmic losses without consideration of computing second derivatives, while the Hessian variant requires computation of second derivatives and leads to greater improvements.

We evaluate Newton Losses for an array of eight families of algorithmic losses on two popular algorithmic benchmarks: the four-digit MNIST sorting benchmark and the Warcraft shortest-path benchmark. We find that Newton Losses leads to performance improvements for each of the algorithms—for some of the algorithms (those which suffer the most from vanishing and exploding gradients) more than doubling the accuracy. In an ablation study, we demonstrate that Newton Losses does not harm training when applied to the convex cross-entropy loss in simple classification settings.

## 2   BACKGROUND & RELATED WORK

The related work comprises algorithmic supervision losses and second-order optimization methods. To the best of our knowledge, this is the first work combining second-order optimization of loss functions with first-order optimization of neural networks, especially for algorithmic losses.

**Algorithmic Losses.**   Algorithmic losses, i.e., losses which contain some kind of algorithmic component, have become quite popular in recent machine learning research. In the domain of recommender systems, early learning-to-rank works already appeared in the 2000s [8], [9], [31], but more recently Lee *et al.* [32] proposed differentiable ranking metrics, and Swezey *et al.* [13] proposed PiRank, which relies on differentiable sorting. For differentiable sorting, an array of methods has been proposed in recent years, which includes NeuralSort [33], SoftSort [34], Optimal Transport Sort [10], differentiable sorting networks (DSN) [12], and the relaxed Bubble Sort algorithm [18]. Other works explore differentiable sorting-based top-k for applications such as differentiable image patch selection [35], differentiable k-nearest-neighbor [17], [33], top-k attention for machine translation [17], differentiable beam search methods [17], [36], survival analysis [37], and self-supervised learning [38]. But algorithmic losses are not limited to sorting: other works have considered learning shortest-paths [15], [16], [18], learning 3D shapes from images and silhouettes [5]–[7], [18], [39], [40], learning with combinatorial solvers for NP-hard problems [16], learning to classify handwritten characters based on editing distances between strings [18], learning with differentiable physics simulations [41], and learning protein structure with a differentiable simulator [42], among many others.

**Second-Order Optimization.**   Second-order methods have gained popularity in machine learning due to their fast convergence properties when compared to first-order methods [19]. One alternative to the vanilla Newton's method are quasi-Newton methods, which, instead of computing an inverse Hessian in the Newton step (which is expensive), approximate this curvature from the change in gradients [26], [43], [44]. In addition, a number of new approximations to the pre-conditioning matrix have been proposed in the recent literature, i.a., [20], [23], [45]. While the vanilla Newton method relies on the Hessian, there are variants which use the empirical Fisher information matrix, which can coincide in specific cases with the Hessian, but generally exhibits somewhat different behavior. For an overview and discussion of Fisher-based methods (aka. natural gradient descent), see [46], [47].

## 3 NEWTON LOSSES

### 3.1 PRELIMINARIES

We consider the training of a neural network $f(x; \theta)$, where $x \in \mathbb{R}^n$ is the vector of inputs, $\theta \in \mathbb{R}^d$ is the vector of trainable parameters and $y = f(x; \theta) \in \mathbb{R}^m$ is the vector of outputs. As per vectorization, $\mathbf{x} = [x_1, \ldots, x_N]^\top \in \mathbb{R}^{N \times n}$ denotes a set of $N$ input data points, and $\mathbf{y} = f(\mathbf{x}; \theta) \in \mathbb{R}^{N \times m}$ denotes the neural network outputs corresponding to the inputs. Further, let $\ell : \mathbb{R}^{N \times m} \to \mathbb{R}$ denote the loss function, and let the "label" information be implicitly encoded in $\ell$. The reason for this choice of implicit notation is that, for many algorithmic losses, it is not just a label, e.g., it can be ordinal information or a set of encoded constraints. The training of a neural network may be expressed as

$$\arg \min_{\theta \in \Theta} \ell(f(\mathbf{x}; \theta)) \tag{1}$$

where $\Theta \subseteq \mathbb{R}^d$ is the domain of the trainable parameters $\theta$. We note that the formulation (1) is general and includes, e.g., optimization of non-decomposable loss functions (i.e., not composed of individual losses per training sample), which is relevant for some algorithmic losses like, e.g., ranking losses [2].

Typically, the optimization problem (1) is solved by using an iterative algorithm (e.g., SGD [48], Adam [49], or Newton's method [26]) updating the weights $\theta$ by repeatedly applying a respective update step:

$$\theta \leftarrow \text{One optimization step of } \ell(f(\mathbf{x}; \theta)) \text{ wrt. } \theta. \tag{2}$$

However, in this work, we consider splitting this optimization update step into two alternating steps:

$$\mathbf{z}^\star \leftarrow \text{One optimization step of } \ell(\mathbf{z}) \text{ wrt. } \mathbf{z} \leftarrow^1 f(\mathbf{x}; \theta), \tag{3a}$$

$$\theta \leftarrow \text{One optimization step of } \tfrac{1}{2} \|\mathbf{z}^\star - f(\mathbf{x}; \theta)\|_2^2 \text{ wrt. } \theta. \tag{3b}$$

This split allows us later to use two different iterative optimization algorithms for (3a) and (3b), respectively. This is especially interesting for optimization problems where the loss function $\ell$ is non-convex and its minimization is a difficult optimization problem itself, and, in other words, those problems, where a more robust and stronger optimization method exhibits faster convergence.

We note that a similar split (for the case of splitting between the layers of a neural network, and using gradient descent for both (3a) and (3b)) is also utilized in the fields of biologically plausible backpropagation [50]–[53] and proximal backpropagation [54], leading to reparameterizations of backpropagation. In Supplementary Material D, we provide further discussion of the split for the special case of using the same optimizer for (3a) and (3b) from a theoretical perspective and show that: (1) for gradient descent the split does not change the parameter updates, (2) for Newton's method in a rank-1 setting the split does not change the parameter updates, and, (3) under simplifying assumptions, the sets of stationary points are invariant to the split for a broader class of deterministic strategies.

### 3.2 METHOD

Equipped with the two-step optimization method (3), we can now introduce Newton Losses:

*We propose optimizing* (3a) *with Newton's method, while optimizing* (3b) *with gradient descent.*

In the following, we formulate how we can solve optimizing (3a) with Newton's method, or, whenever we do not have access to the Hessian of $\ell$, using a step pre-conditioned via the empirical Fisher matrix. This allows us to transform an original loss function $\ell$ into a Newton loss $\ell^*$, which allows optimizing $\ell^*$ with gradient descent, making it suitable for common machine learning frameworks.

We begin by considering the quadratic approximation of the loss function at the location $\mathbf{y} = f(\mathbf{x}; \theta)$

$$\tilde{\ell}_{\mathbf{y}}(\mathbf{z}) = \ell(\mathbf{y}) + (\mathbf{z} - \mathbf{y})^\top \nabla_{\mathbf{y}} \ell(\mathbf{y}) + \tfrac{1}{2}(\mathbf{z} - \mathbf{y})^\top \nabla_{\mathbf{y}}^2 \ell(\mathbf{y}) (\mathbf{z} - \mathbf{y}). \tag{4}$$

To find the location $\mathbf{z}^\star$ of the stationary point of $\tilde{\ell}_{\mathbf{y}}(\mathbf{z})$, we set its derivative to 0:

$$\nabla_{\mathbf{z}} \tilde{\ell}_{\mathbf{y}}(\mathbf{z}^\star) = 0 \iff \nabla_{\mathbf{y}} \ell(\mathbf{y}) + \nabla_{\mathbf{y}}^2 \ell(\mathbf{y})(\mathbf{z}^\star - \mathbf{y}) = 0 \iff \mathbf{z}^\star = \mathbf{y} - (\nabla_{\mathbf{y}}^2 \ell(\mathbf{y}))^{-1} \nabla_{\mathbf{y}} \ell(\mathbf{y}). \tag{5}$$

---

[1]For each step, $\mathbf{z}$ is initialized to the current model predictions $f(\mathbf{x}; \theta)$.

However, when $\ell$ is non-convex or the smallest eigenvalues of $\nabla_{\mathbf{y}}^2\ell(\mathbf{y})$ either become negative or zero, $\mathbf{z}^\star$ would not be the local minimum of $\ell$ but may instead be any other stationary point or lie far away from $\mathbf{y}$, leading to exploding gradients downstream. To resolve this issue, we introduce Tikhonov regularization [55] with a strength of $\lambda$, which leads to a well-conditioned curvature matrix:

$$\mathbf{z}^\star = \mathbf{y} - (\nabla_{\mathbf{y}}^2\ell(\mathbf{y}) + \lambda \cdot \mathbf{I})^{-1}\,\nabla_{\mathbf{y}}\ell(\mathbf{y})\,. \tag{6}$$

Using $\mathbf{z}^\star$, we can plug the solution into (3b) to find the Newton loss $\ell^*$ and compute its derivative as

$$\ell_{\mathbf{z}^\star}^*(\mathbf{y}) = \tfrac{1}{2}(\mathbf{z}^\star - \mathbf{y})^\top(\mathbf{z}^\star - \mathbf{y}) = \tfrac{1}{2}\,\|\mathbf{z}^\star - \mathbf{y}\|_2^2 \qquad \text{and} \qquad \nabla_{\mathbf{y}}\ell_{\mathbf{z}^\star}^*(\mathbf{y}) = \mathbf{y} - \mathbf{z}^\star\,. \tag{7}$$

Via this construction, we obtain the Newton loss $\ell_{\mathbf{z}^\star}^*$, a new convex loss, which itself has a gradient that corresponds to a Newton step of the original loss. Thus, when we optimizing the Newton loss with gradient descent, we obtain the proposed method.

**Definition 1** (Newton Losses (Hessian)). *For a loss function $\ell$ and a given current parameter vector $\theta$, we define the Hessian-based Newton loss via the empirical Hessian as*

$$\ell_{\mathbf{z}^\star}^*(\mathbf{y}) = \tfrac{1}{2}\|\mathbf{z}^\star - \mathbf{y}\|_2^2 \qquad \text{where} \qquad z_i^\star = \bar{y}_i - \left(\tfrac{1}{N}\sum_{j=1}^N \nabla_{\bar{y}_j}^2\ell(\bar{\mathbf{y}}) + \lambda \cdot \mathbf{I}\right)^{-1}\nabla_{\bar{y}_i}\ell(\bar{\mathbf{y}})$$

*for all $i \in \{1, ..., N\}$ and $\bar{\mathbf{y}} = f(\mathbf{x}; \theta)$.*

We remark that computing and inverting the Hessian of the loss function is usually computationally efficient. However, whenever the Hessian matrix of the loss function is not available, whether it may be due to limitations of a differentiable algorithm, large computational cost, lack of a respective implementation of the second derivative, etc., we may resort to using the empirical Fisher matrix as a source for curvature information. We remark that the empirical Fisher matrix is not the same as the Fisher information matrix [46], and that the Fisher information matrix is generally not available for algorithmic losses. While the empirical Fisher matrix, as a source for curvature information, may be of lower quality than the Hessian matrix, it has the crucial advantage that it can be readily computed from the gradients, specifically,

$$\mathbf{F} = \mathbb{E}_{\mathbf{x}}\left[\nabla_f\,\ell(f(\mathbf{x}, \theta)) \cdot \nabla_f\,\ell(f(\mathbf{x}, \theta))^\top\right]\,. \tag{8}$$

This means that, assuming a moderate dimension of the prediction space $m$, computing the empirical Fisher comes at no significant overhead and may, conveniently, be performed in-place as we discuss later. Again, we regularize the matrix via Tikhonov regularization with strength $\lambda$ and can, accordingly, define the empirical Fisher-based Newton loss as follows.

**Definition 2** (Newton Loss (Fisher)). *For a loss function $\ell$, and a given current parameter vector $\theta$, we define the empirical Fisher-based Newton loss as*

$$\ell_{\mathbf{z}^\star}^*(\mathbf{y}) = \tfrac{1}{2}\|\mathbf{z}^\star - \mathbf{y}\|_2^2 \qquad \text{where} \qquad z_i^\star = \bar{y}_i - \left(\tfrac{1}{N}\sum_{j=1}^N \nabla_{\bar{y}_j}\ell(\bar{\mathbf{y}})\,\nabla_{\bar{y}_j}\ell(\bar{\mathbf{y}})^\top + \lambda \cdot \mathbf{I}\right)^{-1}\nabla_{\bar{y}_i}\ell(\bar{\mathbf{y}})$$

*for all $i \in \{1, ..., N\}$ and $\bar{\mathbf{y}} = f(\mathbf{x}; \theta)$.*

Before continuing with the implementation, integration, and further computational considerations, we can make an interesting observation. In the case of using the trivial MSE loss, i.e., $\ell(y) = \tfrac{1}{2}\|y - y^\star\|_2^2$ where $y^\star$ denotes a ground truth, the Newton loss collapses to the original MSE loss. This demonstrates that Newton losses require non-trivial original losses. Another interesting aspect is the arising fixpoint—the Newton loss of a Newton loss is equivalent to a simple Newton loss.

### 3.3 IMPLEMENTATION

After introducing Newton Losses, in this section, we discuss aspects of implementation and integration into differentiable algorithm-based training architectures. In particular, we illustrate its integration in Algorithms 1 and 2. Whenever we have access to the Hessian matrix of the algorithmic loss function, it is generally favorable to utilize the Hessian-based approach (Algo. 1 / Def. 1), whereas we can utilize the empirical Fisher-based approach (Algo. 2 / Def. 2) in any case.

In Algorithm 1, the differences to a regular training is that we use the original `loss` only for the computation of the gradient (`grad`) and the Hessian matrix (`hess`) of the original loss. Then we compute $\mathbf{z}^\star$ (`z_star`). Here, depending on the automatic differentiation framework, we need to ensure

**Algorithm 1** Training with a Newton Loss

```python
# Python style pseudo-code
model = ...        # neural network
loss = ...         # original loss fn
optimizer = ...    # optim. of model
tik_l = ...        # hyperparameter

for data, label in data_loader:
  # apply a neural network model
  y = model(data)

  # compute gradient of orig. loss
  grad = gradient(loss(y, label), y)
  # compute Hessian (or alt. Fisher)
  hess = hessian(loss(y, label), y)

  # compute the projected optimum
  z_star = (y - grad @ inverse(hess
    + tik_l * eye(g.shape[1]))).detach()

  # compute the Newton loss
  l = MSELoss()(y, z_star)

  # backpropagate and optim. step
  l.backward()
  optimizer.step()
```

**Algorithm 2** Training with InjectFisher

```python
# implements the Fisher-based Newton
# loss via an injected modification
# of the backward pass:
class InjectFisher(AutoGradFunction):
  def forward(ctx, x, tik_l):
    assert len(x.shape) == 2
    ctx.tik_l = tik_l
    return x

  def backward(ctx, g):
    fisher = g.T @ g * g.shape[0]
    input_grad = g @ inverse(fisher
        + ctx.tik_l * eye(g.shape[1]))
    return input_grad, None

for data, label in data_loader:
  # apply a neural network model
  y = model(data)
  # inject the Fisher backward mod.
  y = InjectFisher.apply(y, tik_l)
  # compute the original loss
  l = loss(y, label)
  # backpropagate and optim. step
  l.backward()
  optimizer.step()
```

not to backpropagate through the target `z_star`, which may be achieved, e.g., via "`.detach()`", "`.stop_gradient()`", or "`with no_grad():`" depending on the choice of library. Finally, the Newton loss `l` may be computed as a squared or MSE loss between the model output `y` and `z_star` and an optimization step on `l` may be performed.

We note that, while we use a `label` from our `data_loader`, this label may be empty or an abstract piece of information for the differentiable algorithm; in our experiments, we use information about ordinal relationships between values as well as a shortest-path on a graph.

In Algorithm 2, we sketch how to apply the empirical Fisher-based Newton Losses. In particular, due to the empirical Fisher matrix depending only on the gradient, we can compute it in-place during the backward pass / backpropagation, which makes this variant particularly simple and efficient to apply. This can be achieved via an injection of a custom gradient right before applying the original `loss`, which replaces the gradient in-place by a gradient that corresponds to Definition 2. The injection is performed by the `InjectFisher` function, which corresponds to an identity during the forward pass but replaces the gradient by the gradient of the respective empirical Fisher-based Newton loss.

In both cases, the only additional hyperparameter to specify is the Tikonov regularization strength $\lambda$ (`tik_l`). This hyperparameter heavily depends on the algorithmic loss function, particularly, on the magnitude of gradients provided by the algorithmic loss, which may vary drastically between different methods and implementations. Other factors may be the choice of Hessian / Fisher, the dimension of outputs $m$, the batch size $N$. Notably, for very large $\lambda$, the direction of the gradient becomes more similar to regular gradient descent, and for smaller $\lambda$, the effect of Newton Losses increases. We provide an ablation study for $\lambda$ in Figure 4 for the ranking experiment.

## 4 EXPERIMENTS

For the experiments, we apply Newton Losses to eight methods for differentiable algorithms and evaluate them on two established benchmarks for algorithmic supervision, i.e., problems where an algorithm is applied to the predictions of a model and only the outputs of the algorithm are supervised. Specifically, we focus on the tasks of ranking supervision and shortest-path supervision because they each have a range of established methods for evaluating our approach. In ranking supervision, only the relative order of a set of samples is known, while their absolute values remain

unsupervised. The established benchmark for differentiable sorting and ranking algorithms is the multi-digit MNIST sorting benchmark [2], [10], [12], [33], [34]. In shortest path supervision, only the shortest-path of a graph is supervised, while the underlying cost matrix remains unsupervised. The established benchmark for differentiable shortest-path algorithms is the Warcraft shortest-path benchmark [15], [16], [18]. As these tasks require backpropagating through conventionally non-differentiable algorithms, the respective approaches make the ranking or shortest-path algorithms differentiable such that they can be used as part of the loss function. Finally, as an ablation study, we also apply Newton Losses to the trivial case of classification, where we do not expect a performance improvement, as the loss is not hard to optimize, but rather validate that the method still works.

## 4.1 RANKING SUPERVISION

In this section, we explore ranking supervision [33] with an array of differentiable sorting-based losses. Here, we use the four-digit MNIST sorting benchmark [33], where sets of $n$ four-digit MNIST images are given, and the supervision is the relative order of these images corresponding to the displayed value, while the absolute values remain unsupervised. The goal is to learn a CNN that maps each image to a scalar value in an order preserving fashion. As losses, we use sorting supervision losses based on the Neu-

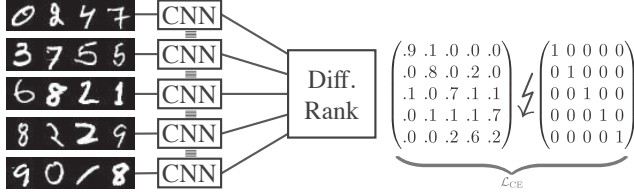

Figure 1: Overview over ranking supervision with a differentiable sorting / ranking algorithm. A set of input images is (element-wise) processed by a CNN, producing a scalar for each image. The scalars are sorted / ranked by the differentiable ranking algorithm, which returns the differentiable permutation matrix, which is compared to the ground truth permutation matrix.

ralSort [33], the SoftSort [34], the logistic Differentiable Sorting Network [12], and the monotonic Cauchy DSN [2]. *NeuralSort* and *SoftSort* work by mapping an input list (or vector) of values to a differentiable permutation matrix that is row-stochastic and indicates the order / ranking of the inputs. *Differentiable Sorting Networks* offer an alternative to NeuralSort and SoftSort. DSNs are based on sorting networks, a classic family of sorting algorithms that operate by conditionally swapping elements. By introducing perturbations, DSNs relax the conditional swap operator to a differentiable conditional swap and thereby continuously relax the sorting and ranking operators. We discuss the background of each of these differentiable sorting and ranking algorithms in greater detail in Supplementary Material A.

**Setups.** The sorting supervision losses are cross-entropy losses defined between the differentiable permutation matrix produced by a respective differentiable sorting operator and the ground truth permutation matrix corresponding to a ground truth ranking. The Cauchy DSN may be an exception to the hard to optimize classification as it is quasi-convex [2]. We evaluate the sorting benchmark for numbers of elements to be ranked $n \in \{5, 10\}$ and use the percentage of rankings correctly identified as well as percentage of individual element ranks correctly identified as evaluation metrics. For each of the four original baseline methods, we compare it to two variants of their Newton losses: the empirical Hessian and the empirical Fisher variant. For each setting, we train the CNN on 10 seeds using the Adam optimizer [49] at a learning rate of $10^{-3}$ for $10^5$ steps and a batch size of 100.

Table 1: Ranking supervision with differentiable sorting. The metric is the percentage of rankings correctly identified (and individual element ranks correctly identified) averaged over 10 seeds. Statistically significant improvements (significance level 0.05) are indicated bold black, and improved means are indicated in bold grey.

| $n = 5$ | NeuralSort [33] | SoftSort [34] | Logistic DSN [12] | Cauchy DSN [2] |
|---|---|---|---|---|
| Baseline | 71.33±2.05 (87.10±0.96) | 70.70±2.60 (86.75±1.26) | 53.56±18.04 (77.04±10.30) | 85.09±0.77 (93.31±0.39) |
| NL (Hessian) | **83.31±1.70** (**92.54±0.73**) | **83.87±0.81** (**92.72±0.39**) | **75.02±12.59** (**88.53±06.00**) | 85.11±0.78 (93.31±0.34) |
| NL (Fisher) | **83.93±0.62** (**92.80±0.30**) | **84.03±0.59** (**92.82±0.24**) | **63.11±30.63** (**79.28±22.16**) | 84.95±0.79 (93.25±0.37) |

| $n = 10$ | NeuralSort | SoftSort | Logistic DSN | Cauchy DSN |
|---|---|---|---|---|
| Baseline | 24.26±01.52 (74.47±0.83) | 27.46±3.58 (76.02±1.92) | 12.31±10.22 (58.81±16.79) | 55.29±2.46 (87.06±0.85) |
| NL (Hessian) | **48.76±05.88** (**84.83±2.13**) | **55.07±1.08** (**86.89±0.31**) | **42.14±22.30** (**75.35±23.77**) | **56.49±1.02** (**87.44±0.40**) |
| NL (Fisher) | **39.23±11.38** (**81.14±4.91**) | **54.00±2.24** (**86.56±0.68**) | **25.72±27.42** (52.18±36.51) | **56.12±1.86** (**87.35±0.65**) |

**Results.** As displayed in Table 1, we can see that—for each original loss—Newton Losses improve over their baselines. For NeuralSort, SoftSort, and Logistic DSNs, we find that using the Newton losses substantially improves performance. Here, the reason is that these methods suffer from vanishing and exploding gradients, especially for the more challenging case of $n = 10$. As expected, we find that the Hessian Newton Loss leads to better results than the Fisher variant, except for NeuralSort and SoftSort in the easy setting of $n = 5$, where the results are nevertheless quite close. Monotonic differentiable sorting networks, i.e., the Cauchy DSNs, provide an improved variant of DSNs, which have the property of quasi-convexity and have been shown to exhibit much better training behavior out-of-the-box, which makes it very hard to improve upon the existing results. Nevertheless, Hessian Newton Losses are on-par for the easy case of $n = 5$ and, notably, improve the performance by more than $1\%$ on the more challenging case of $n = 10$. In summary, we obtain strong improvements on losses that are difficult-to-optimize, while on well-behaving losses the improvements are smaller. This aligns with our goal of improving performance on losses that are hard to optimize.

## 4.2 SHORTEST-PATH SUPERVISION

In this section, we apply Newton Losses to the shortest-path supervision task of the $12 \times 12$ Warcraft shortest-path benchmark [15], [16], [18]. Here, $12 \times 12$ Warcraft terrain maps are given as $96 \times 96$ RGB images (e.g., Figure 2 left) and the supervision is the shortest path from the top left to the bottom right (Figure 2 right) according to a hidden cost embedding (Figure 2 center). The hidden cost embedding is not available for training.

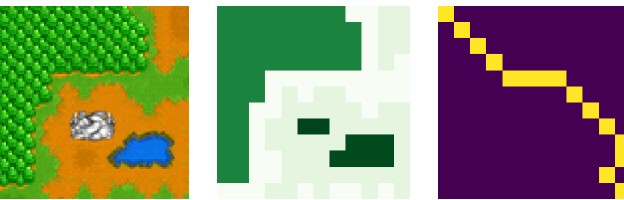

Figure 2: $12 \times 12$ Warcraft shortest-path problem. An input terrain map (left), the unsupervised ground truth cost embedding (center) and the ground truth supervised shortest path (right).

The goal is to predict $12 \times 12$ cost embeddings of the terrain maps such that the shortest path according to the predicted embedding corresponds to the ground truth shortest path. Vlastelica et al. [16] have shown that integrating an algorithm in the training pipeline substantially improves performance compared to only using a neural network with an easy-to-optimize loss function, which has been confirmed by subsequent work [15], [18]. For this task, we explore a set of families of algorithmic supervision approaches: *Relaxed Bellman-Ford* [18] is a shortest-path algorithm relaxed via the AlgoVision framework, which continuously relaxes algorithms by perturbing all accessed variables with logistic distributions and approximating the expectation value in closed form. *Stochastic Smoothing* [56] is a sampling-based differentiation method that can be used to relax, e.g., a shortest-path algorithm by perturbing the input with probability distribution. *Perturbed Optimizers with Fenchel-Young Losses* [15] build on stochastic smoothing and Fenchel-Young losses [57] and identify the argmax to be the differential of max, which allows a simplification of stochastic smoothing, again applied, e.g., to shortest-path learning problems. We use the same hyperparameters as shared by previous works [15], [18].

**Relaxed Bellman-Ford.** The relaxed Bellman-Ford algorithm [18] is a continuous relaxation of the Bellman-Ford algorithm via the AlgoVision library. To increase the number of settings considered, we explore four sub-variants of the algorithm: For+$L_1$, For+$L_2^2$, While+$L_1$, and While+$L_2^2$. Here, For / While refers to the distinction between us-

Table 2: Shortest-path benchmark results for different variants of the AlgoVision-relaxed Bellman-Ford algorithm [18]. The metric is the percentage of perfect matches averaged over 10 seeds. Significant improvements are bold black, and improved means are bold grey.

| Variant | For+$L_1$ | For+$L_2^2$ | While+$L_1$ | While+$L_2^2$ |
|---|---|---|---|---|
| Baseline | 94.19±0.33 | **95.90±0.21** | 94.30±0.20 | 95.77±0.41 |
| NL (Fisher) | **94.52±0.34** | 95.37±0.54 | **94.47±0.34** | **95.93±0.28** |

ing a While and For loop in Bellman-Ford, while $L_1$ vs. $L_2^2$ refer to the choice of metric between shortest paths. As computing the Hessian of the AlgoVision Bellman-Ford algorithm is too expensive with the PyTorch implementation, for this evaluation, we restrict it to the empirical Fisher-based Newton loss. The results displayed in Table 2. While we can observe small improvements in three out out of four settings and the best result is achieved by Newton Losses, the differences are small.

This can be attributed to (i) the high performance of the baseline algorithm on this benchmark, and (ii) only being able to use the empirical Fisher-based Newton loss which is expected to underperform compared to the Hessian Newton loss.

**Stochastic Smoothing.** After discussing the analytical relaxation, we continue with stochastic smoothing approaches. First, we consider stochastic smoothing [56], which allows perturbing the input of a function with an exponential family distribution to estimate the gradient of the smoothed function. For the baseline, we apply stochastic smoothing to a hard non-differentiable Dijkstra algorithm based loss function to relax it via Gaussian noise ("SS of loss"). We utilize variance reduction via the method of covariates. As we detail in Supplementary Material A.4, stochastic smoothing can also be used to estimate the Hessian of the smoothed function [56]. Based on this result, we can construct the Hessian Newton loss. As an extension to stochastic smoothing, we apply stochastic smoothing only to the non-differentiable Dijstra algorithm (thereby computing its Jacobian matrix) but use a differentiable loss to compare the predicted relaxed shortest-path to the ground truth shortest-path ("SS of algorithm"). In this case, the Hessian Newton loss is not applicable because the output of the smoothed algorithm is high dimensional and the Hessian of the loss becomes intractable. Nevertheless, we can apply the Fisher-based Newton loss. We evaluate both approaches for 3, 10, and 30 samples.

In Table 3, we can observe that Newton Losses improves the results for stochastic smoothing in each case with more than 3 samples. The reason for the poor performance on 3 samples is that the Hessian or empirical Fisher, respectively, is estimated using only 3 samples, which makes the estimate unstable. For 10 and 30 samples, the performance improves compared to the original method. In Figure 3, we display a respective accuracy plot. When comparing "SS of loss" and "SS of algorithm", we can observe that the extension to smoothing only the algorithm improves performance for at least 10 samples. Here, the reason, again, is that smoothing the algorithm itself requires estimating the Jacobian instead of only the gradient; thus, a larger number of samples is necessary; however

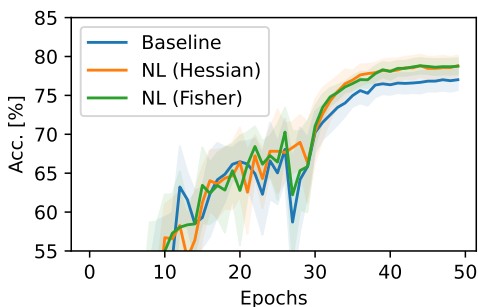

Figure 3: Test accuracy (perfect matches) plot for 'SS of loss' with 10 samples on the Warcraft shortest-path benchmark. Displayed is the mean and 95% conf. interval.

starting at 10 samples, smoothing the algorithm performs better, which means that the approach is better at utilizing a given sample budget.

**Perturbed Optimizers with Fenchel-Young Losses.** Perturbed optimizers with a Fenchel-Young loss [15] is a formulation of solving the shortest path problem as an $\arg\max$ problem, and differentiating this problem using stochastic smoothing-based perturbations and a Fenchel-Young loss. Berthet *et al.* [15] mentioned that their approach works well for small numbers of samples, which we can confirm as seen in Table 3 where the accuracy is similar for each number of samples. By extending their formulation to computing the Hessian of the Fenchel-Young loss, we can compute the Newton loss, and find that we can achieve improvements of more than 2%. However, for Fenchel-Young losses, which are defined via their derivative, the empirical Fisher is not particularly meaningful, leading to equivalent performance between the baseline and the Fisher Newton loss. An interesting observation is that perturbed optimizers with Fenchel-Young losses perform better than stochastic smoothing in the few-sample regime, whereas stochastic smoothing performs better with larger numbers of samples.

Table 3: Shortest-path benchmark results for the stochastic smoothing of the loss (including the algorithm), stochastic smoothing of the algorithm (excluding the loss), and perturbed optimizers with the Fenchel-Young loss. The metric is the percentage of perfect matches averaged over 10 seeds. Significant improvements are bold black, and improved means are bold grey.

| Method | SS of loss | | | SS of algorithm | | | PO w/ FY loss | | |
|---|---|---|---|---|---|---|---|---|---|
| # Samples | 3 | 10 | 30 | 3 | 10 | 30 | 3 | 10 | 30 |
| Baseline | **62.83±5.29** | 77.01±2.18 | 85.48±1.23 | **57.55±4.58** | 78.70±1.90 | 87.26±1.50 | 80.64±0.75 | 80.39±0.57 | 80.71±1.28 |
| NL (Hessian) | 62.40±5.48 | **78.82±2.12** | 85.94±1.33 | — | — | — | **83.09±3.11** | **81.13±3.58** | **83.45±2.21** |
| NL (Fisher) | 58.80±5.10 | **78.74±1.68** | 86.10±0.60 | 53.82±8.45 | **79.24±1.78** | 87.41±1.13 | **80.70±0.65** | 80.37±0.98 | 80.45±0.78 |

### 4.3 Ablation Study

Finally, as an ablation study, we want to ensure that Newton Losses do not impede training for the simple case of MNIST classification with a softmax cross-entropy loss. We remark that, as cross-entropy is convex and very well-behaved objective, we cannot expect improvements for this experiment. To facilitate a fair and extensive comparison, we benchmark training on 5 models and with 2 optimizers: We use 5-layer fully connected ReLU networks with 100 (M1), 400 (M2) and 1 600 (M3) neurons per layer, as well as the convolutional LeNet-5 with sigmoid activations (M4) and LeNet-5 with ReLU activations (M5). Further, we use SGD and Adam [49] as optimizers. To evaluate both early performance and full training performance, we test after 1 and 200 epochs. We run each experiment with 20 seeds, which allows us to perform significance tests (significance level 0.05). As displayed in Table 4, we find that Newton Losses perform indistinguishable to regular training, specifically, in 9 out of 20 cases it is better and in 7 cases equal. We find that it is significantly better in a single case, which is to be expected from equal methods (on average $1/20$ tests will be significant at a significance level of $0.05$). In conclusion, we can observe that for the trivial case of classification with the convex cross-entropy loss, Newton Losses maintains the original training performance.

Table 4: Ablation study: MNIST classification. Results averaged over 20 seeds. A better mean is indicated by a gray bold number and a significantly better mean is indicated by a black bold number.

| | Optim. | SGD Optimizer | | | | | Adam Optimizer | | | | |
|---|---|---|---|---|---|---|---|---|---|---|---|
| Ep. | Method / Model | M1 | M2 | M3 | M4 | M5 | M1 | M2 | M3 | M4 | M5 |
| 1 | SMCE loss | 93.06% | 94.26% | 94.77% | 10.57% | **96.25%** | 94.75% | 96.36% | 95.90% | 90.60% | 97.56% |
| 1 | NL (Hessian) | 93.06% | **94.28%** | 94.77% | 10.57% | 96.23% | 94.75% | 96.36% | **95.91%** | **90.63%** | **97.63%** |
| 200 | SMCE loss | **98.13%** | 98.40% | 98.46% | **99.06%** | 99.07% | 98.12% | 98.46% | 98.62% | 98.95% | **99.23%** |
| 200 | NL (Hessian) | 98.11% | **98.42%** | 98.46% | 99.04% | **99.11%** | 98.12% | **98.50%** | **98.63%** | **98.97%** | 99.20% |

### 4.4 Runtime Analysis

We provide tables with runtimes for the experiments in Supplementary Material B. We can observe that the runtimes between the baseline and empirical Fisher-based Newton Losses are indistinguishable for all cases. For the analytical relaxations of differentiable sorting algorithms, where the computation of the Hessian can become expensive with automatic differentiation (i.e., without a custom derivation of the Hessian and without vectorized Hessian computation), we observed overheads between $10\%$ and $2.6\times$. For all stochastic approaches, we observe indistinguishable runtimes for Hessian-based Newton Losses. In summary, applying the Fisher variant of Newton Losses has a minimal computational overhead, whereas, for the Hessian variant, any overhead depends on the implementation of the computation of the Hessian of the algorithmic loss function. For applications where the the output dimensionality $m$ becomes very large such that inversion of the empirical Fisher becomes expensive, we refer to the Woodbury matrix identity [58], which allows simplifying the computation via its low-rank decomposition. A corresponding deviation is included in Supplementary Material G.

## 5 Conclusion

In this work, we focused on weakly-supervised learning problems that require integration of differentiable algorithmic procedures in the loss function. This leads to non-convex loss functions that exhibit vanishing and exploding gradients, making them hard to optimize. Thus, we proposed a novel approach for improving performance of algorithmic losses building upon the curvature information of the loss. For this, we split the optimization procedure into two steps: optimizing on the loss itself using Newton's method to mitigate vanishing and exploding gradients, and then optimizing the neural network with gradient descent. We simplified this procedure via a transformation of an original loss function into a Newton loss, which comes in two flavors: a Hessian variant for cases where the Hessian is available and an empirical Fisher variant as an alternative. We evaluated Newton Losses on a set of algorithmic supervision settings, demonstrating that the method can drastically improve performance for weakly-performing differentiable algorithms. We hope that the community adapts Newton Losses for learning with differentiable algorithms and see great potential for combining it with future differentiable algorithms in unexplored territories of the space of differentiable relaxations, algorithms, operators, and simulators.

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

## A ALGORITHMIC SUPERVISION LOSSES

In this section, we extend the discussion of SoftSort, DiffSort, AlgoVision, and stochastic smoothing.

### A.1 SoftSort and NeuralSort

SoftSort [34] and NeuralSort [33] are prominent yet simple examples of a differentiable algorithm. In the case of ranking supervision, they obtain an array or vector of scalars and return a row-stochastic matrix called the differentiable permutation matrix $P$, which is a relaxation of the argsort operator. Note that, in this case, a set of $k$ inputs yields a scalar for each image and thereby $y \in \mathbb{R}^k$. As a ground truth label, a ground truth permutation matrix $Q$ is given and the loss between $P$ and $Q$ is the binary cross entropy loss $\ell_{\mathrm{SS}}(y) = \mathrm{BCE}\left(P(y), Q\right)$. Minimizing the loss enforces the order of predictions $y$ to correspond to the true order, which is the training objective. SoftSort is defined as

$$P(y) = \mathrm{softmax}\left(-\left|y^\top \ominus \mathrm{sort}(y)\right|/\tau\right) \tag{9}$$

$$= \mathrm{softmax}\left(-\left|y^\top \ominus Sy\right|/\tau\right) \tag{10}$$

where $\tau$ is a temperature parameter, "sort" sorts the entries of a vector in non-ascending order, $\ominus$ is the element-wise broadcasting subtraction, $|\cdot|$ is the element-wise absolute value, and "softmax" is the row-wise softmax operator, as also used in (32) (right). NeuralSort is defined similarly and omitted for the sake of brevity. In the limit of $\tau \to 0$, SoftSort and NeuralSort converge to the exact ranking permutation matrix [33], [34]. A respective Newton loss can be implemented using automatic differentiation according to Definition 1 or via the Fisher information matrix using Definition 2.

### A.2 DiffSort

Differentiable sorting networks (DSN) [2], [12] offer a strong alternative to SoftSort and NeuralSort. They are based on sorting networks, a classic family of sorting algorithms that operate by conditionally swapping elements [59]. As the locations of the conditional swaps are pre-defined, they are suitable for hardware implementations, which also makes them especially suited for continuous relaxation. By perturbing a conditional swap with a distribution and solving for the expectation under this perturbation in closed-form, we can differentially sort a set of values and obtain a differentiable doubly-stochastic permutation matrix $P$, which can be used via the BCE loss as in Section A.1. We can obtain the respective Newton loss either via the Hessian computed via automatic differentiation or via the Fisher information matrix.

### A.3 AlgoVision

AlgoVision [18] is a framework for continuously relaxing arbitrary simple algorithms by perturbing all accessed variables with logistic distributions. The method approximates the expectation value of the output of the algorithm in closed-form and does not require sampling. For shortest-path supervision, we use a relaxation of the Bellman-Ford algorithm [60], [61] and compare the predicted shortest path with the ground truth shortest path via an MSE loss. The input to the shortest path algorithm is a cost embedding matrix predicted by a neural network.

### A.4 Stochastic Smoothing

Another differentiation method is stochastic smoothing [56]. This method regularizes a non-differentiable and discontinuous loss function $\ell(y)$ by randomly perturbing its input with random noise $\epsilon$ (i.e., $\ell(y + \epsilon)$). The loss function is then approximated as $\ell(y) \approx \ell_\epsilon(y) = \mathbb{E}_\epsilon[\ell(y + \epsilon)]$. While $\ell$ may be non-differentiable, its smoothed stochastic counterpart $\ell_\epsilon$ is differentiable and the corresponding gradient and Hessian can be estimated via the following result.

**Lemma 1** (Exponential Family Smoothing, adapted from Lemma 1.5 in Abernethy *et al.* [56])**.** *Given a distribution over $\mathbb{R}^m$ with a probability density function $\mu$ of the form $\mu(\epsilon) = \exp(-\nu(\epsilon))$ for any twice-differentiable $\nu$, then*

$$\nabla_y l_\epsilon(y) = \nabla_y \mathbb{E}_\epsilon\left[\ell(y + \epsilon)\right] = \mathbb{E}_\epsilon\left[\ell(y + \epsilon)\,\nabla_\epsilon \nu(\epsilon)\right], \tag{11}$$

$$\nabla_y^2 l_\epsilon(y) = \nabla_y^2 \mathbb{E}_\epsilon\left[\ell(y + \epsilon)\right] = \mathbb{E}_\epsilon\left[\ell(y + \epsilon)\left(\nabla_\epsilon \nu(\epsilon)\nabla_\epsilon \nu(\epsilon)^\top - \nabla_\epsilon^2 \nu(\epsilon)\right)\right]. \tag{12}$$

A *variance-reduced form* of (11) and (12) is

$$\nabla_y \mathbb{E}_\epsilon \left[ \ell(y + \epsilon) \right] = \mathbb{E}_\epsilon \left[ \left( \ell(y + \epsilon) - \ell(y) \right) \nabla_\epsilon \nu(\epsilon) \right], \tag{13}$$

$$\nabla_y^2 \mathbb{E}_\epsilon \left[ \ell(y + \epsilon) \right] = \mathbb{E}_\epsilon \left[ \left( \ell(y + \epsilon) - \ell(y) \right) \left( \nabla_\epsilon \nu(\epsilon) \nabla_\epsilon \nu(\epsilon)^\top - \nabla_\epsilon^2 \nu(\epsilon) \right) \right]. \tag{14}$$

In this work, we use this to estimate the gradient of the shortest path algorithm. By including the second derivative, we extend the perturbed optimizer losses to Newton losses. This also lends itself to full second-order optimization.

### A.5 PERTURBED OPTIMIZERS WITH FENCHEL-YOUNG LOSSES

Berthet *et al.* [15] build on stochastic smoothing and Fenchel-Young losses [57] to propose perturbed optimizers with Fenchel-Young losses. For this, they use algorithms, like Dijkstra, to solve optimization problems of the type $\max_{w \in \mathcal{C}} \langle y, w \rangle$, where $\mathcal{C}$ denotes the feasible set, e.g., the set of valid paths. Berthet *et al.* [15] identify the argmax to be the differential of max, which allows a simplification of stochastic smoothing. By identifying similarities to Fenchel-Young losses, they find that the gradient of their loss is

$$\nabla_y \ell(y) = \mathbb{E}_\epsilon \left[ \arg \max_{w \in \mathcal{C}} \langle y + \epsilon, w \rangle \right] - w^\star \tag{15}$$

where $w^\star$ is the ground truth solution of the optimization problem (e.g., shortest path). This formulation allows optimizing the model without the need for computing the actual value of the loss function. Berthet *et al.* [15] find that the number of samples—surprisingly—only has a small impact on performance, such that 3 samples were sufficient in many experiments, and in some cases even a single sample was sufficient. In this work, we confirm this behavior and also compare it to plain stochastic smoothing. We find that for perturbed optimizers, the number of samples barely impacts performance, while for stochastic smoothing more samples always improve performance. If only few samples can be afforded (like 10 or less), perturbed optimizers are better as they are more sample efficient; however, when more samples are available, stochastic smoothing is superior as it can utilize more samples better.

## B RUNTIMES

In this supplementary material, we provide and discuss runtimes for the experiments.

In the differentiable sorting and ranking experiment, as shown in Table 5, we observe that the runtime from regular training compared to the Newton loss with the Fisher is only marginally increased. This is because computing the Fisher and inverting it is very inexpensive. We observe that the Newton loss with the Hessian, however, is more expensive: due to the implementation of the differentiable sorting and ranking operators, we compute the Hessian by differentiating each element of the gradient, which makes this process fairly expensive. An improved implementation could make this process much faster. Nevertheless, there is always some overhead to computing the Hessian compared to the Fisher.

Table 5: Runtimes for the differentiable sorting results corresponding to Table 1. Times of full training in seconds on a single A6000 GPU.

|  | $n = 5$ | | | $n = 10$ | | |
|---|---|---|---|---|---|---|
|  | DSN | NeuralSort | SoftSort | DSN | NeuralSort | SoftSort |
| Baseline | 4214 | 3712 | 3683 | 6192 | 5236 | 5011 |
| NL (Hessian) | 8681 | 4057 | 4216 | 22617 | 6154 | 6010 |
| NL (Fisher) | 4232 | 3762 | 3730 | 6239 | 5153 | 5098 |

In Table 6, we show the runtimes for the shortest-path experiment with AlgoVision. Here, we observe that the runtime overhead is very small.

In Table 7, we show the runtimes for the shortest-path experiment with stochastic methods. Here, we observe that the runtime overhead is also very small. Here, the Hessian is also cheap to compute as it is not computed with automatic differentiation.

Table 6: Runtimes for the shortest-path results corresponding to Table 2. Times of full training in seconds on a single A6000 GPU.

| Algorithm Loop | For | | While | |
|---|---|---|---|---|
| Loss | $L_1$ | $L_2^2$ | $L_1$ | $L_2^2$ |
| Baseline | 624 | 606 | 911 | 913 |
| NL (Fisher) | 608 | 634 | 920 | 917 |

Table 7: Runtimes for the shortest-path results corresponding to Table 3. Times of full training in seconds on a single A6000 GPU.

| Loss | SS of loss | | | SS of algorithm | | | PO w/ FY loss | | |
|---|---|---|---|---|---|---|---|---|---|
| # Samples | 3 | 10 | 30 | 3 | 10 | 30 | 3 | 10 | 30 |
| Baseline | 894 | 1419 | 3183 | 916 | 1401 | 3170 | 646 | 1157 | 2947 |
| NL (Hessian) | 902 | 1398 | 3159 | – | – | – | 666 | 1161 | 2971 |
| NL (Fisher) | 882 | 1394 | 3217 | 901 | 1382 | 3200 | 667 | 1168 | 2973 |

## C  HYPERPARAMETER $\lambda$

For the experiments in the tables, select $\lambda$ based one seed from the grid $\lambda \in [0.001, 0.01, 0.1, 1, 10, 100, 100]$. For the experiments in Tables 2 and 9, we present the values in the Tables 8 and 9, respectively. For the experiment in Table 2, we use a Tikhonov regularization strength of $\lambda = 1000$, which was selected based on the first out of the four algorithms (For+$L_1$).

Table 8: Tikhonov regularization strengths $\lambda$ for the experiment in Table 1.

| | $n = 5$ | | | | $n = 10$ | | | |
|---|---|---|---|---|---|---|---|---|
| | NeuralSort | SoftSort | Logistic DSN | Cauchy DSN | NeuralSort | SoftSort | Logistic DSN | Cauchy DSN |
| NL (Hessian) | $\lambda = 0.01$ | $\lambda = 10$ | $\lambda = 0.1$ | $\lambda = 0.1$ | $\lambda = 0.01$ | $\lambda = 1$ | $\lambda = 0.1$ | $\lambda = 0.1$ |
| NL (Fisher) | $\lambda = 0.1$ | $\lambda = 10$ | $\lambda = 0.1$ | $\lambda = 0.1$ | $\lambda = 100$ | $\lambda = 100$ | $\lambda = 0.1$ | $\lambda = 0.1$ |

Table 9: Tikhonov regularization strengths $\lambda$ for the experiment in Table 3.

| Loss | SS of loss | | | SS of algorithm | | | PO w/ FY loss | | |
|---|---|---|---|---|---|---|---|---|---|
| # Samples | 3 | 10 | 30 | 3 | 10 | 30 | 3 | 10 | 30 |
| NL (Hessian) | $\lambda = 1000$ | $\lambda = 1000$ | $\lambda = 1000$ | — | — | — | $\lambda = 1000$ | $\lambda = 1000$ | $\lambda = 1000$ |
| NL (Fisher) | $\lambda = 0.1$ | $\lambda = 0.1$ | $\lambda = 0.1$ | $\lambda = 1000$ | $\lambda = 1000$ | $\lambda = 1000$ | $\lambda = 1000$ | $\lambda = 1000$ | $\lambda = 1000$ |

In Figure 4, we provide an ablation study for the hyperparameter $\lambda$ for SoftSort and NeuralSort.

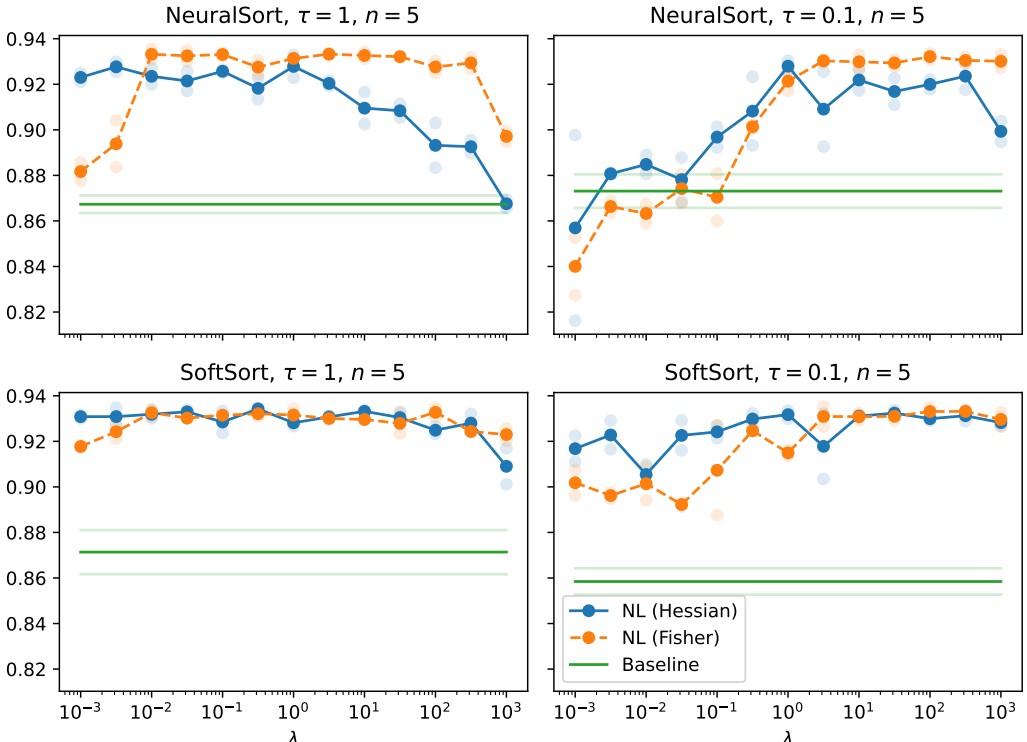

Figure 4: Ablation Study wrt. $\lambda$. Displayed is the accuracy of individual element ranks correctly identified. We use the experimental setting from Table 1 for NeuralSort and SoftSort and $n = 5$ (top right, bottom left). Further, to ablate the hyperparameter of the base methods NeuralSort and SoftSort ($\tau$), we display the results for $\tau \in [1, 0.1]$. Throughout the paper, we selected the optimal hyperparameter settings for each baseline, in particular, $\tau = 0.1$ for NeuralSort and $\tau = 1$ for SoftSort, and used the optimal baseline hyperparameter configurations also for Newton Losses.

In the plots, we consider 13 values for $\lambda$ from $0.001$ to $1000$. We can observe that Newton Losses are robust over multiple orders of magnitude for the hyperparameter $\lambda$.

The plots are averaged over 2 runs and individual runs are indicated with reduced opacity [will be updated to 10 runs for the camera-ready].

# D    EQUIVALENCES UNDER THE SPLIT

In this section, we consider the splitting of the optimization step from (2) into (3a) and (3b) in greater detail. First, we consider the equivalence between (2) and (3a) + (3b) for each case being gradient descent which has also been considered by [54] in a different context. Second, we consider the case of this equivalence for Newton's method and find that it is maintained in a rank-1 setting. Third, we reparameterize optimization steps to show that, under simplifying assumptions, the sets of stationary points are equivalent for a broader class of iterative optimization algorithms.

## D.1    LEMMA 2

Using gradient descent step according to (2) is equivalent to using two gradient steps of the alternating scheme (3), namely one step for (3a) and one step for (3b):

**Lemma 2** (Gradient Descent Step Equality between (2) and (3a)+(3b)). *A gradient descent step according to (2) with arbitrary step size $\eta$ coincides with two gradient descent steps, one according to (3a) and one according to (3b), where the optimization over $\theta$ has a step size of $\eta$ and the optimization over $z$ has a unit step size.*

*Proof.* Let $\theta \in \Theta$ be the current parameter vector and let $\mathbf{z} = f(\mathbf{x}; \theta)$. Then the gradient descent steps according to (3a) and (3b) with step sizes 1 and $\eta > 0$ are expressed as

$$\mathbf{z} \leftarrow \mathbf{z} - \nabla_{\mathbf{z}}\, \ell(\mathbf{z}) = f(\mathbf{x}; \theta) - \nabla_f\, \ell(f(\mathbf{x}; \theta)) \tag{16}$$

$$\theta \leftarrow \theta - \eta\, \nabla_\theta\, \tfrac{1}{2}\|\mathbf{z} - f(\mathbf{x}; \theta)\|_2^2 = \theta - \eta\, \frac{\partial f(\mathbf{x}; \theta)}{\partial \theta} \cdot (f(\mathbf{x}; \theta) - \mathbf{z})\,. \tag{17}$$

Combining (16) and (17) eventually leads to

$$\theta \leftarrow \theta - \eta\, \frac{\partial f(\mathbf{x}; \theta)}{\partial \theta} \cdot (f(\mathbf{x}; \theta) - f(\mathbf{x}; \theta) + \nabla_f\, \ell(f(\mathbf{x}; \theta)))$$
$$= \theta - \eta\, \nabla_\theta\, \ell(f(\mathbf{x}; \theta)), \tag{18}$$

which is exactly a gradient descent step of problem (1) starting at $\theta \in \Theta$ with step size $\eta$. $\qquad\square$

## D.2    LEMMA 3

Here, we show that a corresponding equality also holds for a special case of the Newton step.

**Lemma 3** (Newton Step Equality between (2) and (3a)+(3b) for $m = 1$). *In the case of $m = 1$ (i.e., a one-dimensional output), a Newton step according to (2) with arbitrary step size $\eta$ coincides with two Newton steps, one according to (3a) and one according to (3b), where the optimization over $\theta$ has a step size of $\eta$ and the optimization over $z$ has a unit step size.*

*Proof.* Let $\theta \in \Theta$ be the current parameter vector and let $\mathbf{z} = f(\mathbf{x}; \theta)$. Then applying Newton steps according to (3a) and (3b) leads to

$$\mathbf{z} \leftarrow \mathbf{z} - (\nabla_{\mathbf{z}}^2 \ell(\mathbf{z}))^{-1} \nabla_{\mathbf{z}}\, \ell(\mathbf{z})$$
$$= f(\mathbf{x}; \theta) - (\nabla_f^2 \ell(f(\mathbf{x}; \theta)))^{-1} \nabla_f \ell(f(\mathbf{x}; \theta)) \tag{19}$$

$$\theta \leftarrow \theta - \eta \left( \nabla_\theta^2 \tfrac{1}{2}\|\mathbf{z} - f(\mathbf{x}; \theta)\|_2^2 \right)^{-1} \nabla_\theta \tfrac{1}{2}\|\mathbf{z} - f(\mathbf{x}; \theta)\|_2^2 \tag{20}$$

$$= \theta - \eta \left( \frac{\partial}{\partial \theta} \left[ \frac{\partial f(\mathbf{x}; \theta)}{\partial \theta} \cdot (f(\mathbf{x}; \theta) - \mathbf{z}) \right] \right)^{-1} \frac{\partial f(\mathbf{x}; \theta)}{\partial \theta} \cdot (f(\mathbf{x}; \theta) - \mathbf{z}) \tag{21}$$

$$= \theta - \eta \left( \frac{\partial}{\partial \theta} \left[ \frac{\partial f(\mathbf{x}; \theta)}{\partial \theta} \right] (f(\mathbf{x}; \theta) - \mathbf{z}) + \left( \frac{\partial f(\mathbf{x}; \theta)}{\partial \theta} \right)^2 \right)^{-1} \frac{\partial f(\mathbf{x}; \theta)}{\partial \theta} \cdot (f(\mathbf{x}; \theta) - \mathbf{z})$$

Inserting (19), we can rephrase the update above as

$$\theta \leftarrow \theta - \eta \left( \frac{\partial}{\partial \theta} \left[ \frac{\partial f(\mathbf{x}; \theta)}{\partial \theta} \right] (\nabla_f^2 \ell(f(\mathbf{x}; \theta)))^{-1} \nabla_f \ell(f(\mathbf{x}; \theta)) + \left( \frac{\partial f(\mathbf{x}; \theta)}{\partial \theta} \right)^2 \right)^{-1}$$
$$\cdot \frac{\partial f(\mathbf{x}; \theta)}{\partial \theta} \cdot (\nabla_f^2 \ell(f(\mathbf{x}; \theta)))^{-1} \nabla_f \ell(f(\mathbf{x}; \theta)) \tag{22}$$

By applying the chain rule twice, we further obtain

$$
\begin{aligned}
\nabla_\theta^2 \ell(f(\mathbf{x};\theta)) &= \frac{\partial}{\partial\theta}\left[\frac{\partial f(\mathbf{x};\theta)}{\partial\theta}\nabla_f \ell(f(\mathbf{x};\theta))\right] \\
&= \frac{\partial}{\partial\theta}\left[\frac{\partial f(\mathbf{x};\theta)}{\partial\theta}\right]\nabla_f \ell(f(\mathbf{x};\theta)) + \frac{\partial f(\mathbf{x};\theta)}{\partial\theta}\frac{\partial}{\partial\theta}\nabla_f \ell(f(\mathbf{x};\theta)) \\
&= \frac{\partial}{\partial\theta}\left[\frac{\partial f(\mathbf{x};\theta)}{\partial\theta}\right]\nabla_f \ell(f(\mathbf{x};\theta)) + \frac{\partial f(\mathbf{x};\theta)}{\partial\theta}\nabla_f \frac{\partial}{\partial\theta}\ell(f(\mathbf{x};\theta)) \\
&= \frac{\partial}{\partial\theta}\left[\frac{\partial f(\mathbf{x};\theta)}{\partial\theta}\right]\nabla_f \ell(f(\mathbf{x};\theta)) + \left(\frac{\partial f(\mathbf{x};\theta)}{\partial\theta}\right)^2 \nabla_f^2 \ell(f(\mathbf{x};\theta)),
\end{aligned}
$$

which allows us to rewrite (22) as

$$
\begin{aligned}
\theta' &= \theta - \left((\nabla_f^2 \ell(f(\mathbf{x};\theta)))^{-1}\nabla_\theta^2 \ell(f(\mathbf{x};\theta))\right)^{-1}(\nabla_f^2 \ell(f(\mathbf{x};\theta)))^{-1}\nabla_\theta \ell(f(\mathbf{x};\theta)) \\
&= \theta - (\nabla_\theta^2 \ell(f(\mathbf{x};\theta)))^{-1}\nabla_\theta \ell(f(\mathbf{x};\theta)),
\end{aligned}
$$

which is exactly a single Newton step of problem (1) starting at $\theta \in \Theta$. $\qquad\square$

### D.3 LEMMA 4

In this section, we reparameterize update step of deterministic iterative optimization methods to show an equality of the sets of convergence points under model assumptions. Specifically, we can express optimization steps corr. to (2) and (3a) as

$$
\begin{aligned}
\theta &\leftarrow \arg\min_{\theta'\in\Theta} \ell(f(\mathbf{x};\theta')) + \Omega_1(\theta',\theta) \\
\text{and}\quad \mathbf{z}^\star &\leftarrow \arg\min_{\mathbf{z}\in\mathcal{Y}} \ell(\mathbf{z}) + \Omega_2(\mathbf{z}, f(\mathbf{x};\theta))
\end{aligned}
\tag{23}
$$

where $\Omega$ is a regularizer such that one step of a respective deterministic iterative optimization method (such as gradient descent) corresponds to the global optimum of the regularized optimization problems in (23). More specifically, $\Omega_1$ corresponds to the optimizer used in (2) and $\Omega_2$ corresponds to the optimizer used in (3a). We note that the regularizer $\Omega$ is *not* a neural network regularizer like weight regularization or dropout; instead, the regularizers are reparameterizations of local optimization strategies. The reason for using the perspective of reparameterizing optimization steps as regularized cases of the global optimizations problems is exclusively for enabling a generality of our theory. We further note that a regularizer $\Omega$ is a valid choice if and only if it corresponds to a step of a respective deterministic iterative optimization method. As an example, the regularizer corresponding to a Newton step is $\Omega(\mathbf{z}, f(\mathbf{x};\theta)) = \tilde{\ell}_{f(\mathbf{x};\theta)}(\mathbf{z}) - \ell(\mathbf{z})$. Formally, given an input $\mathbf{x}$ we consider $\mathcal{Y} = \{f(\mathbf{x};\theta)\ ,\ \theta \in \Theta\} \subseteq \mathbb{R}^{N\times m}$ as the set of attainable outputs. Here, we assume $\mathcal{Y} = \mathbb{R}^{N\times m}$, which generally holds for over-parameterized universal function approximators like neural networks, and we assume that such output remains attainable during training, which excludes a catastrophic collapse of the network, e.g., to a constant output. The regularizer $\Omega$ has the standard property that $\Omega(a, b) = 0$ for any $a = b$. Note that the explicit form of the regularizer $\Omega$ does not need to be known nor computed. In SM E, we discuss explicit choices of $\Omega$ and provide further discussion.

This allows us to express the set of points of convergence for the iterative optimization methods. Recall that an iterative optimization method has converged if it has arrived at a fixed point, i.e, the parameters do not change when applying an update. The set of points of convergence for (2) is

$$
\mathcal{A} = \left\{\theta \mid \theta \in \arg\min_{\theta'\in\Theta}\ell(f(\mathbf{x};\theta')) + \Omega_1(\theta',\theta)\right\},
$$

i.e., those points at which the update does not change $\theta$. For the two-step optimization method (3), the set of points of convergence is

$$
\mathcal{B} = \left\{\theta \mid f(\mathbf{x};\theta) = \mathbf{z}^\star \in \arg\min_{\mathbf{z}\in\mathcal{Y}}\ell(\mathbf{z}) + \Omega_2(\mathbf{z}, f(\mathbf{x};\theta))\right\}
$$

as the method has converged if the update (3a) yields $\mathbf{z}^\star = \mathbf{z}$ because the subsequent update (3b) will not change $\theta$ as $\mathbf{z}^\star = \mathbf{z} = f(\mathbf{x};\theta)$ already holds, and thus $\frac{1}{2}\|\mathbf{z}^\star - f(\mathbf{x};\theta)\|_2^2 = 0$.

Now, we show that the iterative method (2) and the alternating method (3) lead to the same sets of convergence points, i.e., to the same sets possible solutions.

**Lemma 4** (Equality of the Sets of Convergence Points). *Assuming $\Omega_1 = \Omega_2$ being valid in accordance to Equation 23, and assuming the model as described in this section, the set $\mathcal{A}$ of points of convergence obtained by the iterative optimization method (2) is equal to the set $\mathcal{B}$ of points of convergence obtained by the two-step iterative optimization method (3).*

*Proof.* ($\mathcal{A} \subset \mathcal{B}$)    First, we show that any point in $\mathcal{A}$ also lies in $\mathcal{B}$. By definition, for each point in $\mathcal{A}$, the optimization step (2) does not change $\theta$, i.e., $\theta' = \theta$. Thus, $f(\mathbf{x}; \theta) = f(\mathbf{x}; \theta') \in \arg\min_{\mathbf{z} \in \mathcal{Y}} \ell(\mathbf{z}) + \Omega_2(\mathbf{z}, f(\mathbf{x}; \theta))$, and therefore $\theta \in \mathcal{B}$.

($\mathcal{B} \subset \mathcal{A}$)    Second, we show that any point in $\mathcal{B}$ also lies in $\mathcal{A}$. For each $\theta \in \mathcal{B}$, we know that, by definition, $f(\mathbf{x}; \theta) = \mathbf{z}^\star \in \arg\min_{\mathbf{z} \in \mathcal{Y}} \ell(\mathbf{z}) + \Omega_2(\mathbf{z}, f(\mathbf{x}; \theta))$, therefore $\theta \in \arg\min_{\theta' \in \Theta} \ell(f(\mathbf{x}; \theta')) + \Omega_2(f(\mathbf{x}; \theta'), f(\mathbf{x}; \theta))$ where $\Omega_2(f(\mathbf{x}; \theta), f(\mathbf{x}; \theta)) = 0 = \Omega_1(\theta, \theta)$, and, therefore $\theta \in \mathcal{A}$.  $\square$

Lemma 4 states the equivalence of the original training (2) and its counterpart (3) wrt. their possible points of convergence (i.e., solutions) for an arbitrary choice of the iterative method as long as $\Omega_1 = \Omega_2$, i.e., as long as the optimization method in (2) equals the optimization method in (3a).

**Remark 1** (Generality of Lemma 4). *The choice of the optimization method in (3b) is arbitrary and does not have to match the one used in (3a). Thus, Lemma 4 states that the points of convergence (i.e., solutions) are the same between using Newton's method and using Newton Losses.*

## E    REGULARIZERS $\Omega$

In this section, we provide a more detailed discussion, how the regularization term $\Omega$ in (23) induces different iterative optimization methods.

We note that simply choosing any regularizer that does not correspond to a respective optimization step, such as $\Omega_\neg(x, y) = \|x - y\|_2^2$, is not valid. The reason for this is that we require a *regularizer that is a reparameterization of a respective optimization step*.

The simple setting, where (3a) represents a basic gradient descent, i.e.,

$$\mathbf{z}^\star \leftarrow \mathbf{z} - \eta \nabla \ell(\mathbf{z}), \quad \mathbf{z} = f(\mathbf{x}; \theta), \tag{24}$$

can be obtained by choosing a regularization term as

$$\Omega(\mathbf{z}, f(\mathbf{x}; \theta)) = \ell(f(\mathbf{x}; \theta)) + \eta \nabla_f \ell(f(\mathbf{x}; \theta))^\top (\mathbf{z} - f(\mathbf{x}; \theta))$$
$$+ \tfrac{1}{2}(\mathbf{z} - f(\mathbf{x}; \theta))^\top (\mathbf{z} - f(\mathbf{x}; \theta)) - \ell(\mathbf{z}). \tag{25}$$

Then, the first-order optimality conditions for $\min_{\mathbf{z}} \ell(\mathbf{z}) + \Omega(\mathbf{z}, f(\mathbf{x}; \theta))$ are

$$\nabla_z (\ell(\mathbf{z}) + \Omega(\mathbf{z}, f(\mathbf{x}; \theta))) = \eta \nabla_f \ell(f(\mathbf{x}; \theta)) + \mathbf{z} - f(\mathbf{x}; \theta) = 0, \tag{26}$$

which leads to the gradient step

$$\mathbf{z} = f(\mathbf{x}; \theta) - \eta \nabla_f \ell(f(\mathbf{x}; \theta)). \tag{27}$$

Alternatively, we can ask for the choice of the regularization term $\Omega$ corresponding to a Newton step in (3a), i.e.,

$$\mathbf{z}^\star \leftarrow \mathbf{z} - \eta (\nabla^2 \ell(\mathbf{z}))^{-1} \nabla \ell(\mathbf{z}), \quad \mathbf{z} = f(\mathbf{x}; \theta). \tag{28}$$

In this case, by choosing

$$\Omega(\mathbf{z}, f(\mathbf{x}; \theta)) = \ell(f(\mathbf{x}; \theta)) + \eta(\mathbf{z} - f(\mathbf{x}; \theta))^\top \nabla_f \ell(f(\mathbf{x}; \theta)) \tag{29}$$
$$+ \tfrac{1}{2}(\mathbf{z} - f(\mathbf{x}; \theta))^\top \nabla_f^2 \ell(f(\mathbf{x}; \theta))(\mathbf{z} - f(\mathbf{x}; \theta)) - \ell(\mathbf{z}),$$

the first-order optimality conditions for $\min_{\mathbf{z}} \ell(\mathbf{z}) + \Omega(\mathbf{z}, f(\mathbf{x}; \theta))$ are

$$\nabla_z (\ell(\mathbf{z}) + \Omega(\mathbf{z}, f(\mathbf{x}; \theta))) = \eta \nabla_f \ell(f(\mathbf{x}; \theta)) + \nabla_f^2 \ell(f(\mathbf{x}; \theta))(\mathbf{z} - f(\mathbf{x}; \theta)) = 0, \tag{30}$$

which is equivalent to a Newton step

$$\mathbf{z} = f(\mathbf{x}; \theta) - \eta (\nabla^2 \ell(f(\mathbf{x}; \theta)))^{-1} \nabla \ell(f(\mathbf{x}; \theta)). \tag{31}$$

## F    NEWTON LOSSES OF TRIVIAL LOSSES

For illustrative purposes, in this section, we present examples of Newton losses of trivial losses, which we can solve in closed form.

A popular loss function for classification is the softmax cross entropy (SMCE) loss, defined as

$$\ell_{\text{SMCE}}(y) = \sum_{i=1}^{k} -p_i \log q_i \,, \tag{32}$$

$$\text{where} \quad q_i = \frac{\exp(y_i)}{\sum_{j=1}^{k} \exp(y_j)} \,.$$

**Example 1** (Softmax cross-entropy loss). *For the SMCE loss, the induced Newton loss is given as*

$$\ell_{\text{SMCE}}^{*}(y) = \frac{1}{2} \| z^{\star} - y \|_2^2 \tag{33}$$

*where the element-wise Hessian variant is*

$$z_E^{\star} = - \left( \text{diag}(q) - qq^{\top} \right)^{-1} (q - p) + y \,, \tag{34}$$

*the empirical Hessian variant is*

$$z_H^{\star} = -\mathbb{E} \left[ \text{diag}(q) - qq^{\top} \right]^{-1} (q - p) + y \,, \tag{35}$$

*and the empirical Fisher variant is*

$$z_F^{\star} = -\mathbb{E} \left[ (q - p)(q - p)^{\top} \right]^{-1} (q - p) + y \,. \tag{36}$$

A less trivial example is the binary cross-entropy (BCE) loss with

$$\ell_{\text{BCE}}(y) = \text{BCE}(y, p) \tag{37}$$

$$= - \sum_{i=1}^{m} p_i \log y_i + (1 - p_i) \log(1 - y_i) \,,$$

where $p \in \Delta_m$ is a probability vector encoding of the ground truth.

**Example 2** (Binary cross-entropy loss). *For the BCE loss, the induced Newton loss is given as*

$$\ell_{\text{BCE}}^{*}(y) = \frac{1}{2} \| z^{\star} - y \|_2^2 \,, \tag{38}$$

*where the element-wise Hessian variant is*

$$z_E^{\star} = - \text{diag} \left( -p \oslash y^2 + (1 - p) \oslash (1 - y)^2 \right)^{-1}$$
$$(p \oslash y - (1 - p) \oslash (1 - y)) + y, \tag{39}$$

*the empirical Hessian variant is*

$$z_H^{\star} = - \text{diag} \left( \mathbb{E}_y \left[ -p \oslash y^2 + (1 - p) \oslash (1 - y)^2 \right] \right)^{-1}$$
$$(p \oslash y - (1 - p) \oslash (1 - y)) + y, \tag{40}$$

*and the empirical Fisher variant is*

$$z_F^{\star} = -\mathbb{E}_y \left[ (p \oslash y - (1 - p) \oslash (1 - y)) \right.$$
$$\left. (p \oslash y - (1 - p) \oslash (1 - y))^{\top} \right]^{-1}$$
$$(p \oslash y - (1 - p) \oslash (1 - y)) + y, \tag{41}$$

*and where $\odot$ and $\oslash$ are element-wise operations.*

The BCE loss is often extended using the logistic sigmoid function to what is called the sigmoid binary cross-entropy loss (SBCE), defined as

$$\ell_{\text{SBCE}}(y) = \text{BCE}(\sigma(y), p) \tag{42}$$

$$\text{where} \quad \sigma(x) = \frac{1}{1 + \exp(-x)} \,.$$

**Example 3** (Sigmoid Binary cross-entropy loss). *For the SBCE loss, the induced Newton loss is given as*

$$\ell^*_{\text{SBCE}}(y) = \frac{1}{2} \left\| z^\star - y \right\|_2^2 \tag{43}$$

*where the element-wise Hessian variant is*

$$z^\star_E = -\operatorname{diag}\left(\sigma(y) - \sigma(y)^2\right)^{-1}(\sigma(y) - p) + y\,, \tag{44}$$

*the empirical Hessian variant is*

$$z^\star_H = -\operatorname{diag}\left(\mathbb{E}_y\left[\sigma(y) - \sigma(y)^2\right]\right)^{-1}(\sigma(y) - p) + y\,, \tag{45}$$

*and the empirical Fisher variant is*

$$z^\star_F = -\mathbb{E}_y\left[(\sigma(y) - p)(\sigma(y) - p)^\top\right]^{-1}(\sigma(y) - p) + y\,, \tag{46}$$

*and where $\odot$ and $\oslash$ are element-wise operations.*

## G  WOODBURY MATRIX IDENTITY FOR HIGH DIMENSIONAL OUTPUTS

For the empirical Fisher method, where $\mathbf{z}^\star \in \mathbb{R}^{N \times m}$ is computed via

$$z^\star_i = \bar{y}_i - \left(\frac{1}{N}\sum_{j=1}^{N}\nabla_{\bar{y}_j}\ell(\bar{\mathbf{y}})\,\nabla_{\bar{y}_j}\ell(\bar{\mathbf{y}})^\top + \lambda \cdot \mathbf{I}\right)^{-1}\nabla_{\bar{y}_i}\ell(\bar{\mathbf{y}})$$

for all $i \in \{1, ..., N\}$ and $\bar{\mathbf{y}} = f(\mathbf{x}; \theta)$, we can simplify the computation via the Woodbury matrix identity [58]. In particular, we can simplify the computation of the inverse to

$$\left(\frac{1}{N}\sum_{j=1}^{N}\nabla_{\bar{y}_j}\ell(\bar{\mathbf{y}})\,\nabla_{\bar{y}_j}\ell(\bar{\mathbf{y}})^\top + \lambda \cdot \mathbf{I}\right)^{-1} \tag{47}$$

$$= \left(\frac{1}{N}\nabla_{\bar{\mathbf{y}}}\ell(\bar{\mathbf{y}})^\top\,\nabla_{\bar{\mathbf{y}}}\ell(\bar{\mathbf{y}}) + \lambda \cdot \mathbf{I}\right)^{-1} \tag{48}$$

$$= \frac{1}{\lambda} \cdot \mathbf{I}_m - \frac{1}{N \cdot \lambda^2} \cdot \nabla_{\bar{\mathbf{y}}}\ell(\bar{\mathbf{y}})^\top \cdot \left(\frac{1}{N \cdot \lambda}\nabla_{\bar{\mathbf{y}}}\ell(\bar{\mathbf{y}})\,\nabla_{\bar{\mathbf{y}}}\ell(\bar{\mathbf{y}})^\top + \mathbf{I}_N\right)^{-1} \cdot \nabla_{\bar{\mathbf{y}}}\ell(\bar{\mathbf{y}})\,. \tag{49}$$

This reduces the cost of matrix inversion from $\mathcal{O}(m^3)$ down to $\mathcal{O}(N^3)$. We remark that this variant is only helpful for cases where the batch size $N$ is smaller than the output dimensionality $m$.

Further, we remark that in a case where the output dimensionality $m$ and the batch size $N$ are both very large, i.e., the cost of the inverse is typically still small in comparison to the computational cost of the neural network. To illustrate this, the dimensionality of the last hidden space (i.e., before the last layer) $h_l$ is typically larger than the output dimensionality $m$. Accordingly, the cost of only the last layer itself is $\mathcal{O}(N \cdot m \cdot h_l)$, which is typically larger than the cost of the inversion in Newton Losses. In particular, assuming $h_l > m$, $\mathcal{O}(N \cdot m \cdot h_l) > \mathcal{O}(\min(m, N)^3)$.

We remark that the Woodbury inverted variation can also be used in the `InjectFisher` implementation.

