# OpenReview forum: "Newton Losses: Using Curvature Information for Learning with Differentiable Algorithms"
_ICLR.cc/2024/Conference — Submitted to ICLR 2024_

### Official Review · Reviewer_x7SJ · 2023-10-27

**Soundness:** 3 good
**Presentation:** 4 excellent
**Contribution:** 2 fair
**Rating:** 6
**Confidence:** 3

**Summary:**

This is an interesting paper about split method with quadratic approximation.

**Strengths:**

well written and good presentation. It also has solid theories support.

**Weaknesses:**

There are some missing details and comparisons in the experiments part.

**Questions:**

Here are some questions:
Q1: Is there training curve of the real loss functions L during model training? Is there mismatch or conflicts between the MSE loss of the quadratic approximation and the real training loss for the real output y?
Q2: what is the training time for this new method, compared with original SGD method? Ex, the number of iterations and total training time. the validation test accuracy curve during training.

---

> ### Author Response · Authors · 2023-11-21
>
> We thank the reviewer for evaluating our work. In the following, we address your comments.
>
> Thank you for appreciating our "well written and good presentation" and our "solid theories support".
>
> > "Q1: Is there training curve of the real loss functions L during model training? Is there mismatch or conflicts between the MSE loss of the quadratic approximation and the real training loss for the real output y?"
>
> The Newton Loss indeed has a different absolute loss value compared to the respective original loss, which as-per-construction of the Newton Loss does not pose a conflict. Further, we note that the original loss value is still available during training as the Newton Losses build upon the original loss.
> Figure 3 shows the test accuracy during training and we have included axis labels in the revised PDF.
>
> > "Q2: what is the training time for this new method, compared with original SGD method? Ex, the number of iterations and total training time. the validation test accuracy curve during training."
>
> Kindly see Supplementary Material B where we show runtimes for the experiments. In particular, it shows that the proposal has comparable runtime in most cases and only where the computation of the Hessian is slow, it can slow down training. Therefore, we also proposed training with the Fisher variant, which always has comparable runtime compared to the original SGD method.
>
> > "There are some missing details and comparisons in the experiments part."
>
> Please let us know whether we resolved your all of your concerns or whether there are any other details you would like us to elaborate on. If we successfully addressed your concerns, we would appreciate if you would consider upgrading your score.

---

### Official Review · Reviewer_gz7t · 2023-10-28

**Soundness:** 3 good
**Presentation:** 3 good
**Contribution:** 3 good
**Rating:** 6
**Confidence:** 5

**Summary:**

The authors study new algorithms for weakly-supervised learning problems that involve differentiable algorithmic procedures within the loss function. These problems often result in non-convex loss functions with vanishing and exploding gradients, making them difficult to optimize. The authors propose a novel two-step approach to tackle this issue:

1. They first optimize the loss function using Newton's method to handle the gradient issues.
2. Then, they optimize the neural network parameters with gradient descent. The key innovation is the introduction of "Newton Losses," which come in two variants: one using the Hessian (if available) and another using an empirical Fisher variant when the Hessian is not available. The authors tested these Newton Losses in various algorithmic settings and demonstrated significant performance improvements for weakly-performing differentiable algorithms.

Numerical experiments demonstrate the effectiveness of the method.

**Strengths:**

The method proposes to study a natural gradient method, true one or approximate one in terms of samples, in solving supervised learning problems. Numerical experiments demonstrate the advantage of the proposed method.

**Weaknesses:**

1. There are very few analytical examples studied in the paper. This could help readers realize the importance of Fisher information matrix.

2. There are general natural gradient methods in the literature, in particular the Wasserstein natural gradient method. This could be discussed in the introduction.

Li, et.al. Natural gradient via optimal transport.

Li et.al. Affine Natural Proximal Learning.

**Questions:**

A suitable theortical study is needed for the proposed paper. Some analytical examples can be presented to demonstrate the effectiveness of proposed method.

---

> ### Author Response · Authors · 2023-11-21
>
> We thank the reviewer for evaluating our work. In the following, we address your comments.
>
> > "There are very few analytical examples studied in the paper. This could help readers realize the importance of Fisher information matrix.
> >
> >  A suitable theortical study is needed for the proposed paper. Some analytical examples can be presented to demonstrate the effectiveness of proposed method."
>
> It would be helpful if the reviewer could point us to what is meant with "analytical examples".
> In particular, we would like kindly to point the reviewer to Supplementary Materials A, D, E and F, where we study the method from a theoretical perspective. For example, in SM D, we provide a variety of theoretical statements, and in SM F we illustrate analytically how the method would be applied to trivial losses.
>
> > "There are general natural gradient methods in the literature, in particular the Wasserstein natural gradient method. This could be discussed in the introduction.
> >
> > Li, et.al. Natural gradient via optimal transport.
> > Li et.al. Affine Natural Proximal Learning."
>
> We thank the reviewer for pointing us to these two references. Natural gradients are not directly applicable to differentiable algorithms (because they require particular structures of the loss, not given in the case of differentiable algorithm based losses). It is still very interesting to see that natural gradients have also been considered in combination with optimal transport, and extending these ideas to be applicable for differentiable algorithms could be interesting for future investigations. We have included them as suggested in the introduction.
>
> Please let us know whether we resolved your all of your concerns or whether there are any specific analytical examples you would like us to elaborate on. If we successfully addressed your concerns, we would appreciate if you would consider upgrading your score.

---

> > ### Comment · Reviewer_gz7t · 2023-11-21
> > **Reply to authors**
> >
> > Thanks for answering my questions. The authors address these concerns. I increased my score from 5 to 6.

---

### Official Review · Reviewer_RBot · 2023-10-30

**Soundness:** 3 good
**Presentation:** 3 good
**Contribution:** 3 good
**Rating:** 6
**Confidence:** 4

**Summary:**

This paper proposes Newton Losses, a new optimization scheme for training machine learning models with hard-to-optimize losses. The idea is to compute second-order updates of the loss with respect to the model output, and then back-propagating those to the parameters using standard first-order methods. This can be viewed as replacing the original loss with a new quadratic loss around the minimizer of a quadratic approximation of the original loss, which the authors refer to as a Newton Loss.

Two possible second-order methods are presented for approximating the original loss around a batch of model outputs, based on either the Hessian or the empirical Fisher, leading to two possible instantiations of Newton losses. Both methods are regularized with Tikhonov regularization, which controls the influence of the Newton loss compared to standard gradient descent.

The implementation is shown to be simple for the Hessian Newton Loss, which requires accessing and inverting the Hessian, and even simpler for the Fisher Newton Loss, which only requires minor modifications to the implementation of standard gradient descent.
Two weakly-supervised problem settings are considered, namely MNIST digit sorting and Warcraft shortest paths. In these settings 8 algorithmic losses are compared to their Newton loss counterparts. Both the Fisher and the Hessian Newton Loss are shown to yield improved performance, with the margin depending on the difficulty of the algorithmic loss function as well as the quality of the Hessian approximation.

**Strengths:**

This paper proposes a novel method for optimizing machine learning models with algorithmic losses. The paper was a good read, the method and results are presented with clarity. The experimental setup and the underlying choices were described in detail, and the improvements seem to be consistent over the setups. Good explanations were also provided for the cases in which the method does not work well, which gives a nice guideline for practitioners that want to employ this method. The authors have also been transparent about potential limitations regarding the runtime.

**Weaknesses:**

While the results show consistent improvements in the experiments, the improvement upon the best performing baselines seems relatively minor compared to the observed standard deviations.
However, the method seems to especially improve performance on suboptimal baselines, which could transfer very well to potential new experimental setups.

Regarding the presentation, I think the standard deviations should also be included in the main paper and not deferred to the appendix, as they seem very important for correctly interpreting the presented results. I also have some additional questions below. If the authors adequately address these final concerns I believe this paper will make a good contribution to the conference.

**Questions:**

- In Tables 8-10 it looks like a wide range of the parameter $\lambda$ seems to be used in the experiments. Have the authors found these values purely empirically, or have they been chosen based on e.g. gradient norm? It would also be a useful addition to include an ablation study of how sensitive the performance is to this hyper-parameter for one of the experimental setups.
- In figure 3, does the plot show the number of training epochs on the x-Axis? It seems like this information is missing. This plot could also benefit from averaging over all seeds and including the standard deviations.
- In the proof of Lemma 3, does getting from equation 22 to the final expression require some assumptions on commutativity of the involved matrices? It would be great if the authors could present this final step in a bit more detail.

---

> ### Author Response · Authors · 2023-11-21
>
> We thank the reviewer for evaluating our work. In the following, we address your comments.
>
> **Weaknesses**
>
> > "While the results show consistent improvements in the experiments, the improvement upon the best performing baselines seems relatively minor compared to the observed standard deviations. However, the method seems to especially improve performance on suboptimal baselines, which could transfer very well to potential new experimental setups."
>
> Indeed, our method leads to the greatest improvements for the earlier differentiable algorithms. We selected these two categories (sorting and shortest path) of differentiable algorithms explicitly because they each have a number of algorithms that improved upon each other to paint a full picture of the approach. Our motivation is indeed to provide the approach for other researchers to use for future differentiable algorithm setups, which are typically not as well optimized as, e.g., the recent Cauchy DSNs.
>
> > "Regarding the presentation, I think the standard deviations should also be included in the main paper and not deferred to the appendix, as they seem very important for correctly interpreting the presented results. I also have some additional questions below. If the authors adequately address these final concerns I believe this paper will make a good contribution to the conference."
>
> We would like to offer to include statistical significance tests for Tables 1, 2, and 3 for the camera-ready of the paper, considering the space restrictions. We believe that this would paint an even clearer picture than just including the raw standard deviations. Nevertheless, if you still believe that we should also include the standard deviations in exchange for a smaller table font size, we would accomodate that.
>
> **Questions**
>
> > "In Tables 8-10 it looks like a wide range of the parameter $\lambda$ seems to be used in the experiments. Have the authors found these values purely empirically, or have they been chosen based on e.g. gradient norm? It would also be a useful addition to include an ablation study of how sensitive the performance is to this hyper-parameter for one of the experimental setups."
>
> We found these values empirically by considering the grid $\lambda\in[0.001, 0.01, 0.1, 1, 10, 100, 1000]$ for 1 seed (different seed from the seeds used for the tables).
> We have started an ablation study to show the effects of $\lambda$ on the differentiable sorting experiment over the full grid. As these experiments will not finish within the rebuttal period, we offer to include the ablation study for the camera-ready.
>
> > "In figure 3, does the plot show the number of training epochs on the x-Axis? It seems like this information is missing. This plot could also benefit from averaging over all seeds and including the standard deviations."
>
> Yes, it is the number of epochs. We added labels and extended the figure to include 95% confidence intervals.
>
> > "In the proof of Lemma 3, does getting from equation 22 to the final expression require some assumptions on commutativity of the involved matrices? It would be great if the authors could present this final step in a bit more detail."
>
> In Lemma 3, we particularly consider the case of $m=1$ (i.e., a one-dimensional output) (see first line of Lemma 3).
> With $f(x;\theta)$ a scalar (and hence $\nabla_f$ and $\nabla_f^2$ also scalars), commutativity is not an issue, because scalars commute with matrices.
>
> Please let us know whether we resolved your all of your concerns or whether you have any other questions, concerns, or comments. If we successfully addressed your concerns, we would appreciate if you would consider updating your score.

---

> > ### Comment · Reviewer_RBot · 2023-11-22
> > **Answer to rebuttal**
> >
> > Thank you for responding to my questions. I believe the $\lambda$ ablation will be a nice addition to the paper, I understand this takes longer than the rebuttal period and would like to see it in the camera ready. Regarding the standard deviations, I think they should be added to the main text in exchange for a smaller font. A statistical significance test in the appendix on top of that would of course be nice, but I am not sure if it is necessary on top of the standard deviations. My misunderstanding of the proof of Lemma 3 has also been resolved.
> >
> > Besides this I would also like to comment on the review of reviewer osWY and the authors answer to it. Although I understand that the dimensionality of $y$ is small compared to the model parameters, I share the concern that (at least for the applicability of the method to future settings with higher dimensional $y$) explicitly forming the inverse may be a significant limitation. I also think reviewer osWY already provided some really good ideas for potentially resolving this limitation by directly solving the sparse linear system for $z^*_i$ in Definition 2 without forming the inverse directly. I highly recommend at least mentioning this limitation along with the potential solution, to guide users dealing with high-dimensional $y$.

---

> > > ### Author Response · Authors · 2023-11-23
> > > **Thank you very much for responding in detail to our rebuttal**
> > >
> > > Thank you very much for responding in detail to our rebuttal. We have now updated the PDF again, in particular:
> > >
> > > * We have now included the standard deviations in the experiment tables, and have indicated significant improvements visually.
> > >
> > > * Further, the second seed (out of 10 seeds) for the $\lambda$ ablation study just finished.
> > > Accordingly, we have now included a preliminary plot (Figure 4), which we will update for the camera-ready.
> > > If space for the camera-ready permits, we offer to include one of the subplots in the paper.
> > >
> > > * Moreover, we have now included a short discussion of the case of large dimensionality $m$ in Section 4.4 and have also included the corresponding derivation for efficient computation of this case in Supplementary Material G.
> > >
> > > Please let us know whether we successfully addressed all of your concerns or whether you have any remaining question, remarks, or concerns.

---

### Official Review · Reviewer_osWY · 2023-10-30

**Soundness:** 3 good
**Presentation:** 3 good
**Contribution:** 2 fair
**Rating:** 5
**Confidence:** 3

**Summary:**

The presented study considers the problem of training weakly-supervised problems, including differentiable shortest path or sorting algorithms. The main idea is to split the computing of the loss function in a superposition of two functions and update the corresponding parameters in an alternating manner. In particular, Newton method update is used for the first function and the standard SGD-like update is used for the second one. Also, such splitting at least preserves stationary points of the initial loss function. In Newton's method, both the exact Hessian matrix and the Fisher matrix are considered to precondition the gradient.  The authors provide pseudocode on how to incorporate the proposed approach in PyTorch-like code. The experiments demonstrate that Newton Losses lead to higher test accuracy for sorting supervision losses and shortest-path supervision.

**Strengths:**

The strengths of the presented study are listed below
- the main text is well-written. The motivation, appropriate context, and the proposed approach are clear and original.
- the construction of Newton's losses has a theoretical basis which guarantees the same stationary points
- experimental results are solid and demonstrate the performance of the proposed approach over the baseline

**Weaknesses:**

The main weakness of the presented approach is that the authors consider explicit construction of Hessian or Fischer matrices and directly inverse it. Matrix inversion is the numerically unstable operation in the ill-conditioned case, see (Higham, 1993 - https://eprints.maths.manchester.ac.uk/346/1/covered/MIMS_ep2006_167.pdf). This issue may be crucial in the single-precision format which is typically used in deep learning. In addition, storage of the full square matrix with a dimension equal to the number of model parameters can be prohibited for large models. Thus, this step in the proposed method lacks detailed analysis and requires more in-depth investigation in terms of scalability and robustness to round-off errors. For example, the iterative methods for solving large symmetrix linear systems can exploit hessian-by-vector product operation without explicitly composing the hessian matrix due to automatic differentiation tools. Also, the linear systems in the optimizing first function can be solved with different tolerances in the Inexact Newton method manner, which may significantly affect both runtime and final performance.

**Questions:**

1) how scalable is your approach in terms of the size of the dataset and the number of model parameters?
2) what are specific issues related to the solving of weakly supervised learning problems? The authors mention non-convexity and exploding/vanishing gradients, but the same issues arise in the classical image classification models. Please, provide more details regarding the mentioned learning problems' features.
3) the presented runtime analysis in Supplementary B confuses me since in some cases the proposed method with additional costs per iteration is faster than the baseline. Please provide a more detailed description of the reason for such a result. Also, I disagree that the inversion of the matrix is inexpensive since it is $O(n^3)$ complexity and it is small only for the small dimension $n$. Therefore, we again come to question 1) on the scalability of the proposed method
4) the authors report only mean values of the target metrics in the main text. Could you please append the standard deviations, too, for a fair comparison of the methods?
5) please, add axis labels in Fig. 3
6) also, captions to tables are typically placed above the table, not below

---

> ### Author Response · Authors · 2023-11-21
>
> We thank the reviewer for evaluating our work. In the following, we address your comments.
>
> **Weaknesses:**
>
> > "The main weakness of the presented approach is that the authors consider explicit construction of Hessian or Fischer matrices and directly inverse it. Matrix inversion is the numerically unstable operation in the ill-conditioned case, see (Higham, 1993 - https://eprints.maths.manchester.ac.uk/346/1/covered/MIMS_ep2006_167.pdf). This issue may be crucial in the single-precision format which is typically used in deep learning."
>
> We use Tikhonov regularization on the Hessian and Fisher matrices, which is a standard approach to improve their conditioning by "shifting the eigenvalues" (as the condition number of a matrix is the ratio of the largest to the smallest eigenvalue) (this is addressed in the paper from Equations (5) to (6)). Numerical problems are thus avoided. Note that the Fisher matrix, as an outer product, is necessarily positive semi-definite, and with Tikhonov regularization guaranteed to be positive definite.
>
> > "In addition, storage of the full square matrix with a dimension equal to the number of model parameters can be prohibited for large models. Thus, this step in the proposed method lacks detailed analysis and requires more in-depth investigation in terms of scalability and robustness to round-off errors. For example, the iterative methods for solving large symmetrix linear systems can exploit hessian-by-vector product operation without explicitly composing the hessian matrix due to automatic differentiation tools. [...]"
>
> Indeed a "full square matrix with a dimension equal to the number of model parameters" could be prohibitively expensive.
> However, in our work, we never consider such a matrix. Instead the Hessian or Fisher matrices are always of size $m\times m$ where **$m$ is the dimensionality of the output of the neural network** (see Section 3.1, all Hessian / Fisher matrices are wrt. $\mathbf{z}$, i.e., the output of the neural network).
> To illustrate this, for a regular classification neural network, $m$ would correspond to the number of classes (often, 10--1000), which is always drastically smaller than the number of parameters (often millions to billions). In our paper, we consider sizes of $m$ in the range of 5 to 144.
>
> We believe that many of the questions build on this misunderstanding.
>
> **Questions:**
>
> 1. The approach is scalable regardless of the number of model parameters and the size of the dataset. Scalability only depends on the number of outputs of the neural networks / number of inputs of the differentiable algorithm / loss. Another component can be the cost of computing the Hessian of a differentiable algorithm; however, for this reason, we also include the Fisher approach.
>
> 2. In this work, we deal with non-convexity and exploding/vanishing gradients specifically in the loss function (and not the neural network). In classical image classification, the loss is typically convex and does not have exploding/vanishing gradients (e.g., CrossEntropyLoss). Conceptually, we are dealing with problems where optimization of the loss function is the bottleneck of the learning problem and optimizing the neural network is easier than optimizing the loss. In image classification with convex losses, this is not the case.
>
> 3. Regarding the runtime analysis, the largest matrix that we need to invert is a well-conditioned 144x144 matrix, which is computationally very cheap (< 0.1 millisecond). Accordingly, in most experiments the runtime is only insignificantly increased, and as many factors affect the runtime (like GPU temperature, GPU die quality, CPU scheduling, PCIe interferences, etc.) the differences between runtimes are often smaller than the possible measurement precision of the runtimes. The only case where the runtime actually gets noticably more expensive was in the case of differentiable sorting with the Hessian because computing the Hessian of the differentiable sorting algorithms did not have closed-form solutions (using the Fisher resolves this concern.)
>
> 4. The standard deviations are reported in the Supplementary Material C. The sizes of the standard deviations are actually common in the field, which is why we followed the gold standard of using 10 seeds. We found that most improvements are actually statistically significant and we offer to indicate statistical significances in the camera-ready of the paper.
>
> 5. Thanks for the suggestion, we added the axis labels to Fig. 3
>
> 6. Thanks, we moved the captions on top of the tables.
>
> Please let us know whether we resolved your all of your concerns or whether you have any other questions, concerns, or comments. If we successfully addressed your concerns, we would appreciate if you would consider updating your score.

---

### Meta-Review · Area_Chair_mcqd · 2023-12-02

**Metareview:**

The paper considers an interesting problem of solving weakly-supervised learning problems involving algorithmic loss functions. This is done by forming a quadratic approximation of the loss, and splitting the optimization step into two alternating steps. The paper provide numerical examples on several problems to showcase the advantages of their approach. Although the paper is well-written and contains reasonable numerical experiments, the underlying idea of quadratic approximation of the loss and splitting the iterations appears to be marginal contribution, lacking substantial novelty. Although the paper introduces some theoretical aspects, the assumptions upon which they rely are overly simplifying and do not align well with realistic settings.

**Justification For Why Not Higher Score:**

The underlying idea of quadratic approximation of the loss and splitting the iterations is a rather marginal contribution and lacks substantial novelty. The theoretical justifications rely on assumptions that are overly simplifying and do not align well with realistic settings.

**Justification For Why Not Lower Score:**

N/A

---

### Decision · Program_Chairs · 2024-01-16

Reject